# Analysis of Local Recurrence Risk in Ductal Carcinoma In Situ and External Validation of the Memorial Sloan Kettering Cancer Center Nomogram

**DOI:** 10.3390/cancers15082392

**Published:** 2023-04-21

**Authors:** Gabriela Oses, Eduard Mension, Claudia Pumarola, Helena Castillo, León Francesc, Inés Torras, Isaac Cebrecos, Xavier Caparrós, Sergi Ganau, Belén Ubeda, Xavier Bargallo, Blanca González, Esther Sanfeliu, Sergi Vidal-Sicart, Reinaldo Moreno, Montserrat Muñoz, Gorane Santamaría, Meritxell Mollà

**Affiliations:** 1Department of Radiation Oncology, Hospital Clínic of Barcelona, 08036 Barcelona, Spain; 2Department of Obstetrics and Gynecology, Hospital Clínic of Barcelona, 08036 Barcelona, Spain; 3Department of Radiology, Hospital Clínic of Barcelona, 08036 Barcelona, Spain; 4Departament of Pathology, Hospital Clínic of Barcelona, 08036 Barcelona, Spain; 5Departament of Nuclear Medicine, Hospital Clínic of Barcelona, 08036 Barcelona, Spain; 6Department of Medical Oncology, Hospital Clínic of Barcelona, 08036 Barcelona, Spain; 7Department of Radiology, Princess Alexandra Hospital, Brisbane 4102, Australia

**Keywords:** ductal carcinoma in situ, risk factors of local recurrence, MSKCC nomogram

## Abstract

**Simple Summary:**

Adjuvant treatments after breast-conserving surgery (BCS) in ductal carcinoma in situ (DCIS) have shown to reduce the risk of ipsilateral breast tumor relapse (IBTR), but its effect on overall survival remains controversial. Due to this argument, some nomograms tried to select the patients who may benefit the most to these not free of side-effects adjuvant treatments. In this study, an external validation of the Memorial Sloan Kettering Cancer Center (MSKCC) nomogram was performed not reaching statistical significance in a Spanish cohort of patients. Nonetheless, the expression of estrogen receptors, not included in the MSKCC nomogram, showed potential prediction power for IBTR in the studied population and may be considered in future nomograms.

**Abstract:**

Background: Adjuvant radiotherapy and hormonotherapy after breast-conserving surgery (BCS) in ductal carcinoma in situ (DCIS) have been shown to reduce the risk of local recurrence. To predict the risk of ipsilateral breast tumor relapse (IBTR) after BCS, the Memorial Sloan Kettering Cancer Center (MSKCC) developed a nomogram to analyze local recurrence (LR) risk in our cohort and to assess its external validation. Methods: A historical cohort study using data from 296 patients treated for DCIS at the Hospital Clínic of Barcelona was carried out. Patients who had had a mastectomy were excluded from the analysis. Results: The mean age was 58 years (42–75), and the median follow-up time was 10.64 years. The overall local relapse rate was 13.04% (27 patients) during the study period. Actuarial 5- and 10-year IBTR rates were 5.8 and 12.9%, respectively. The external validation of the MSKCC nomogram was performed using a multivariate logistic regression analysis on a total of 207 patients, which did not reach statistical significance in the studied population for predicting LR (*p* = 0.10). The expression of estrogen receptors was significantly associated with a decreased risk of LR (OR: 0.25; *p* = 0.004). Conclusions: In our series, the LR rate was 13.4%, which was in accordance with the published series. The MSKCC nomogram did not accurately predict the IBTR in this Spanish cohort of patients treated for DCIS (*p* = 0.10).

## 1. Introduction

Ductal carcinoma in situ (DCIS) is a proliferation of malignant cells limited to ductal units that do not invade through the basement membrane and, as such, are theoretically incapable of spreading to regional lymph nodes or metastasizing. DCIS comprises approximately 20% of mammographically detected breast cancer cases [1], but if relapse after local treatment for DCIS occurs, 50% of recurrences present as an invasive ductal carcinoma, which shortens the life of these patients [2,3,4].

Four randomized trials and a meta-analysis have already proven that whole-breast radiotherapy (WBRT) after breast-conserving surgery (BCS) for DCIS reduced the risk of ipsilateral breast tumor relapse (IBTR) by 40 to 60% [2,3,4,5,6,7]. The National Surgical Adjuvant Breast and Bowel Project B-17 trial and other metanalyses confirmed that all DCIS subgroups benefited from postoperative WBRT [3,4,5], which decreased IBTR. As a result of these data, BCS followed by WBRT has become the standard treatment for all patients presenting with DCIS, even though WBRT did not lead to a significant increase in overall survival [4].

Similarly, endocrine therapy (ET) for DCIS has also been proven to reduce the incidence of IBTR and contralateral tumors, but it has had no effect on overall survival [4,5,6,7,8]. In an NSABP B-24 trial, patients presenting with DCIS and positive estrogen receptor (ER) benefitted from tamoxifen [9]. Since then, the National Comprehensive Cancer Network (NCCN) has recommended ER-staining of DCIS to guide adjuvant ET recommendations [10], while progesterone receptor (PR) and HER2 status are currently not routinely used for DCIS decision making.

Nevertheless, considering the lack of effect that WBRT and ET had on overall survival, some authors suggested that a subgroup of patients may benefit from omitting these therapies and that the potential advantage of these adjuvant treatments has to be balanced against potential morbidity [11]. Moreover, to select which patients should receive adjuvant therapies for DCIS remains controversial. The Van Nuys Prognostic Index (VNPI) was developed to aid clinical decision making by stratifying IBTR risk according to nuclear grade, the presence of necrosis, margin width, and later age, to choose between excision alone, excision with radiation therapy, or mastectomy [12,13]. However, the validity of VNPI remains open to discussion due to conflicting reports from external data sets [14,15,16,17].

Afterwards, the Memorial Sloan Kettering Cancer Center (MSKCC) nomogram was developed to assist physicians in making decisions about the various treatment options to avoid the over- or undertreatment of noninvasive breast cancer [18]. This DCIS nomogram integrates 10 clinicopathologic variables to provide an individualized risk estimate of IBTR in a woman with DCIS who received BCS.

The aim of this study was to analyze the local recurrence (LR) risk in a Spanish cohort and assess the external validation of the MSKCC nomogram to predict the risk of ipsilateral breast tumor relapse (IBTR) after BCS.

## 2. Methods

### 2.1. Design and Subjects

A historical cohort study was carried out using data from 296 patients treated for DCIS at the Hospital Clínic of Barcelona between 1999 and 2019. Thirty percent (89) of the patients had had a mastectomy and were excluded from the analysis (Figure 1).

This study was approved by the Ethics Committee of the Hospital Clínic of Barcelona, Spain (HCB/2022/0791), and all the patients gave their written informed consent.

The main outcome assessed among patients was the ipsilateral breast tumor relapse (IBTR). The following risk factors for IBTR were analyzed: age at diagnosis, family history, clinical presentation, nuclear grade, focality, comedonecrosis, final surgical resection margin, and the number of reinterventions and adjuvant treatments with radiotherapy and hormone therapy. These are the variables included in the MSKCC nomogram. Additionally, other risk factors were assessed in our cohort: estrogen and progesterone receptors, type of surgery, and pathological tumor size.

Family history was considered positive if a first-degree relative had been diagnosed with breast cancer. Presentation with clinical findings included women with DCIS detected by palpable mass. Margin width was classified as positive/close (≤2 mm) or negative (>2 mm); nuclear grade as low (1) or intermediate/high (2/3); focality as unifocal or multifocal; and comedonecrosis as present or absent. The positivity of the expression of estrogen and progesterone receptors was defined as a value ≥ 10%.

In our cohort, the standard of treatment with DCIS after local excision was WBRT. However, patients with small, low-grade tumors presenting wide margins of excision could choose to avoid it [2,3,4,5,6,7], and some declined WBRT despite the clinical recommendation. Endocrine therapy was advised for patients presenting with ER-positive DCIS.

### 2.2. Statistical Analysis

Statistical analyses were performed using the Software for Statistics and Data Science release 15.1 (STATA: StataCorp LLC, College Station, TX, USA). Descriptive analyses of quantitative variables were made and presented as a mean ± standard deviation. Continuous variables were compared using Student’s *t*-test, and Kaplan–Meier curves were used to evaluate IBTR. Statistically significant differences among risk factors were evaluated independently using logistic regression. Odds ratios (ORs) from the univariate analysis of possible prognostic variables were assessed, and their 95% confidence intervals (CIs) were reported. To validate the predictive MSKCC model, a multivariate logistic regression using the model’s risk factors was performed. The odds ratios and their 95% CIs from this analysis were reported separately: *p* < 0.05 was considered statistically significant.

Finally, two internal validations of the MSKCC nomogram were conducted using bootstrap resampling and cross-validation. Hundreds of samples from the original sample were extracted and used to calculate the logistic model and provide an area-under-the-curve (AUC) analysis of the predictive model.

## 3. Results

Table 1 shows the cohort demographics. The mean age was 58 years (42–75) and the median follow-up time was 10.64 years (min 1 year–max 21 years). In total, 85% (174) of the patients underwent treatment with WBRT; the remaining 15% did not because the tumor was low-risk (small tumor, a low nuclear grade, correct resection margin) or because they refused despite the clinical recommendation. Only 59 patients (28%) adhered correctly to endocrine treatment despite the fact that 74% presented with positive estrogen receptors. The clinicopathologic characteristics of the entire population (*n* = 207) are presented in Table 1.

The IBTR rate was 13.04% (27/207) during the study period: 48.15% (13/27) of patients experienced DCIS recurrence, and 51.85% (14/27) experienced recurrence with invasive breast cancer. Actuarial 5- and 10-year IBTR rates for all patients were 5.8 and 12.9%, respectively.

Table 2 presents the results of the logistical regression analysis of risk factors for the 207 patients, but these did not make a significant difference to IBTR risk: age under 50 at diagnosis, family history, clinical presentation (in the form of a nodule), nuclear grade 2/3, pathological tumor size greater than 20 mm, presence of multifocality and comedonecrosis, non-expression of progesterone receptors, final resection margin less than 2 mm, more reinterventions, and the omission of adjuvant radiotherapy. However, the non-expression of estrogen receptors was significantly associated with a higher risk of LR (*p* = 0.004). The omission of adjuvant endocrine therapy presented a slight but insignificant tendency (OR 0.39) toward an increased risk of IBTR.

To perform the external validation of the MSKCC nomogram, a multivariate logistical regression was performed using the following variables: age, family history, clinical presentation, radiotherapy, endocrine therapy, nuclear grade, presence of comedonecrosis, non-expression of progesterone receptors, final resection margin, and number of reinterventions. No significant difference (*p* = 0.10) was reached in our study cohort for predicting IBTR risk using the nomogram variables in the multivariate logistical regression analysis.

A resampled bootstrap analysis that obtained a cross-validation of 1000 replications was used to validate the MSKCC nomogram internally in our patients’ cohort; obtain an area-under-the-curve (AUC) of 0.7268212; a bootstrap standard error = 0.0622864; and a 95% CI = 0.6047422–0.8489002.

Regarding patient survival, by the end of the study, 93.2% (193/207) were alive, 0.9% (2/207) had died of breast cancer, and 5.8% (12/207) died of other causes.

## 4. Discussion

This study evaluated DCIS relapses in a Spanish cohort of women and showed a 13.04% risk of IBTR. The main finding of this study is that the MSKCC nomogram did not accurately predict the IBTR in patients treated for DCIS. According to our data, the only risk factor that showed a statistically significant relation to the relapse was the absence of estrogen receptors expression, which is actually not a variable included in the assessed MSKCC nomogram.

DCIS is a heterogeneous disease with a large number of clinical and pathologic variables that can influence the risk of IBTR [19]. These include age [20,21,22], clinical presentation [21,22,23], family history [24], multifocality [25], size [13], margin status [26], volume of disease at closest margin [27], histopathologic features such as nuclear grade [28], estrogen and progesterone receptors, presence of necrosis, and architectural pattern [5,29], but these factors alone cannot predict the likelihood of local recurrence risk.

Nomograms are predictive models, based on statistics, that provide the overall probability of a specific outcome (IBTR) for an individual patient. In 2010, investigators from the Memorial Sloan Kettering Cancer Center (MSKCC) published a nomogram that predicted the 5- and 10-year individual probability of local relapse after breast-conserving surgery for DCIS. In this study, 1681 patients treated with BCS for DCIS were analyzed. The median follow-up period was 5.6 years. An IBTR occurred in 202 patients, 122 experienced a recurrence that presented with DCIS, and 80 presented with invasive breast cancer. Actuarial 5- and 10-year IBTR rates for all patients were 9 and 15%, respectively [18]. The MSKCC nomogram presented the results of the multivariate analysis for risk factors: age, family history, clinical presentation, radiotherapy, endocrine therapy, nuclear grade, presence of comedonecrosis, non-expression of progesterone receptors, final resection margin, and number of reinterventions.

However, this nomogram was only internally validated from 200 bootstrap samples, with an estimated concordance probability of 0.668; therefore, applicability to the external data sets was unclear. In addition, the inclusion of tumor size and the endocrine and molecular variables, which may have reflected tumor biology better, were not included [30,31,32]. For these reasons, we decided to test the value of the nomogram with our population to provide independent validation.

In our study, after the multivariate analysis of the MSKCC nomogram risk factors, no significant differences were obtained regarding the risk of IBRT (*p* = 0.10). Only the omission of adjuvant ET presented a slight but insignificant tendency (OR 0.39) to an increased risk of IBTR.

To perform external validation, the MD Anderson Cancer Center (MDACC) performed an external validation with the MSKCC nomogram (734 patients with a DCIS who had undergone local excision). The follow-up time in the MDACC cohort was longer than in the MSKCC cohort (median 7.1 vs. 5.6 years), and the recurrence rate was lower in the MDACC cohort (7.9 vs. 11%). Among the 794 patients, 63 (7.9%) developed IBTR; 57% of IBTRs were presented as an invasive disease (with or without DCIS), and 42.9% as DCIS. The IBTR rate was 4.7% at 5 years and 10.4% at 10 years. After the multivariable analyses, only the omission of ET and the initial presentation on the clinical exam in this cohort were significantly associated with an increased risk of IBTR, demonstrating that the nomogram did not accurately predict the IBTR [32].

Moreover, another external validation of the nomogram was carried out in an Asian population. A cohort of 716 patients presenting with DCIS was analyzed and had a median follow-up of 5.8 years. In total, 42 patients (5.9%) developed IBTR, and for 18 of them (42.9%), it was a recurrent invasive disease, whereas in the remaining 24 patients (57.1%), it recurred as DCIS. In the multivariate analysis of the external MSKCC nomogram validation, these three factors were found to affect IBTR risk significantly: non-use of adjuvant ET, presence of necrosis, and younger age at diagnosis [33].

Finally, an external validation of the MSKCC nomogram was carried out in 467 patients with DCIS at the University Hospital Leuven with a median follow-up of 7.2 years. IBTR was present in 48 (10.3%) patients, of whom 22 developed a pre-invasive lesion, and 26 had an invasive breast cancer relapse. Actuarial 5- and 10-year IBTR rates were 7.0 and 12.5%, respectively. Multivariable analyses of this group of patients were performed. The omission of adjuvant ET, younger age at diagnosis, and positive or close surgical margins were significantly associated with an increased risk of IBTR. The MSKCC nomogram in these series of patients was found to be externally valid [34].

According to reviewed studies, the MSKCC nomogram remains controversial due to the variability in the results among different centers [31,32,33,34]. When assessing it on a Spanish cohort of patients treated for DCIS, it did not present statistical significance. These results are in accordance with other studies assessing the external validation of the nomogram, such as the one presented by MDACC and Wang et al.

Interestingly, the present study assessed a relation to IBTR not only for the MSKCC nomogram-included variables, but other possible risk factor variables discarded by the MSKCC. As other authors have already suggested [35], among the risk factors assessed, those patients with a negative estrogen receptor had a significantly increased risk of IBTR. The lack of important molecular markers was the major weakness of the actual DCIS nomograms; therefore, according to different external validations of the MSKCC nomogram, it is probably not a very strong or reproducible prognostic tool. Further nomograms including molecular markers such as estrogen receptor positivity should be assessed in the future to provide a stronger DCIS prognostic tool.

## 5. Limitations

The number of patients was one of the limitations of our study, despite the fact that it is one of the longest follow-up studies evaluating DCIS recurrence. Another limitation was that we only used clinical and pathological factors to predict the probability of IBTR.

## 6. Conclusions

In our series, the local recurrence rate was 13.4%, which was in accordance with the published series. The MSKCC nomogram external validation in the present cohort did not reach statistical significance. Regarding other potential predictive IBTR risk factors, non-expression of estrogen receptors was significantly associated with an increased risk of local recurrence.

New molecular analyses in the diagnosis of DCIS and the combination of molecular biology and clinicopathological risk factors may define DCIS with a low and high risk of IBTR better and help clinicians individualize DCIS management.

## Figures and Tables

**Figure 1 cancers-15-02392-f001:**
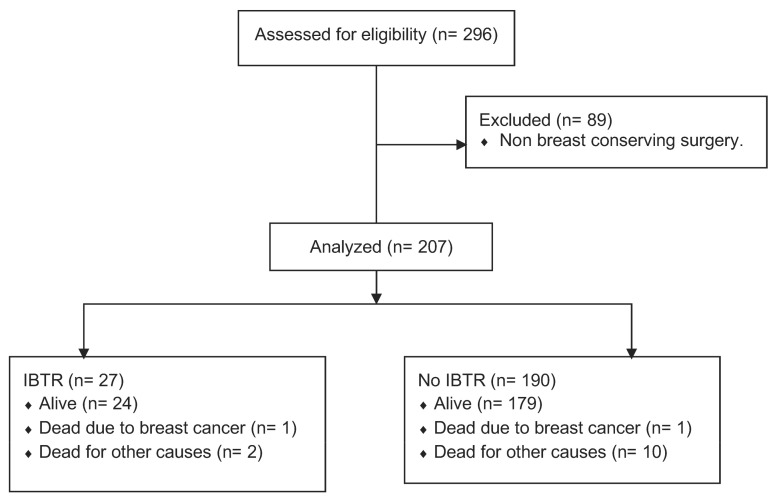
Historical cohort flowchart.

**Table 1 cancers-15-02392-t001:** The clinicopathologic characteristics of patients with DCIS treated with BCS.

Risk Factors	*n* = 207 (%)
Age	
<50 years	27 (13)
>50 years	180 (87)
Family history	
No	107 (52)
Yes	15 (7)
Unknown	85 (41)
Clinical presentation	
No	194 (93.7)
Yes	13 (6.3)
Nuclear grade	
1	68 (32.8)
2–3	130 (62.8)
Unknown	9 (4.4)
Pathological tumor size	
<20 mm	142 (68.5)
>20 mm	62 (30)
Unknown	3 (1.5)
Focality	
Unifocal	176 (85)
Multifocal	31 (15)
Comedonecrosis	
No	90 (43.5)
Yes	114 (55)
Unknown	3 (1.5)
Estrogen receptors	
Positive	154 (74.4)
Negative	35 (16.9)
Unknown	18 (8.7)
Progesterone receptors	
Positive	108 (52.2)
Negative	73 (35.3)
Unknown	26 (12.5)
Final surgical resection margin	
>2 mm	166 (80.2)
<2 mm	40 (19.3)
Unknown	1 (0.5)
Number of reinterventions	
1	181 (87.5)
2	23 (11)
3	1 (0.5)
Unknown	2 (1)
Radiotherapy	
No	33 (15)
Yes	174 (85)
Hormonotherapy	
No	148 (71.5)
Yes	59 (28.5)

**Table 2 cancers-15-02392-t002:** Univariate and multivariate analysis for risk factors of IBRT.

Risk Factors	No IBRT (*n* = 180)	IBRT (*n* = 27)	Univariate OR and CI (95%)	Multivariate OR and CI (95%)	*p*
Included in MSKCC:					
Age					
<50 years	23 (12.8)	4 (14.8)	0.9 (0.9–1.0)	1.0 (0.9–1.1)	0.54
>50 years	157(87.2)	23 (85.2)			
Family history					
No	86 (47.7)	21 (77.8)			
Yes	13 (7.2)	2 (7.4)	0.6 (0.1–3.0)	0.7 (0.1–3.6)	0.54
Unknown	81 (45.0)	4 (14.8)			
Clinical presentation					
No	171 (95.0)	23 (85.2)	3.3 (0.9–11.6)	7.0 (0.9–52.0)	0.08
Yes	9 (5.0)	4 (14.8)			
Radiotherapy					
No	28 (15.6)	5 (18.5)	0.8 (0.3–2.8)	1.3 (0.3–6.2)	0.69
Yes	152 (84.4)	22 (81.5)			
Endocrine Therapy					
No	125 (69.4)	23 (85.1)			
Yes	55 (30.6)	4 (14.8)	0.4 (0.1–1.1)	0.4 (0.9–1.4)	0.07
Nuclear grade					
1	59 (32.8)	9 (33.3)			
2–3	115 (63.8)	15 (55.5)	0.9 (0.3–2.1)	1.2 (0.4–3.7)	0.69
Unknown	6 (3.3)	3 (11.1)			
Comedonecrosis					
No	78 (43.3)	12 (44.4)			
Yes	100 (55.5)	14 (51.8)	0.9 (0.4–2.1)	0.7 (0.2–2.2)	0.82
Unknown	2 (1)	1 (3.7)			
Final surgical resection margin					
>2 mm	147 (81.7)	19 (70.3)			
<2 mm	33 (18.3)	7 (25.9)	1.6 (0.6–4.2)	1.8 (0.5–6.0)	0.31
Unknown	0 (0)	1 (3.7)			
Number of reinterventions					
1	159 (88.8)	22 (81.4)			
2	19 (10.6)	4 (14.8)	1.5 (0.5–4.8)	1.6 (0.5–6.0)	0.49
3	1 (0.0)	0 (0)			
Unknown	1 (0.0)	1 (3.7)			
Other Risk Factors:					
Pathological tumor size					
<20 mm	122 (67.8)	20 (74.1)			
>20 mm	56 (31.1)	6 (22.2)	0.6 (0.2–1.7)		0.28
Unknown	2 (1.1)	1 (3.7)			
Focality					
Unifocal	153 (85.0)	23 (85.2)	1.0 (0.3–3.0)		0.97
Multifocal	27 (15.0)	4 (14.8)			
Estrogen receptors					
Positive	140 (77.8)	14 (51.8)			
Negative	25 (13.9)	10 (37.0)	0.3 (0.1–0.6)		0.004
Unknown	15 (8.3)	3 (11.1)			
Progesterone receptors					
Positive	95 (52.8)	13 (48.1)			
Negative	62 (34.4)	11 (40.7)	0.7 (0.3–1.8)		0.55
Unknown	23 (12.8)	3 (11.1)			

## Data Availability

All data were collected in an electronic database and managed in accordance with privacy regulations. Mension E. and Oses G. had full access to all the data in the study and take responsibility for the integrity of the data and the accuracy of the data analysis. Data and materials will be shared when requested on individual demand to the corresponding author.

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
