# Peer review of "Analysis of Local Recurrence Risk in Ductal Carcinoma In Situ and External Validation of the Memorial Sloan Kettering Cancer Center Nomogram"

_cancers, 2023, doi:10.3390/cancers15082392_

Round 1

Reviewer 1 Report

Authors interrogated their single-institution database of DCIS patients treated with breast conserving surgery to see if variables in the MSKCC nomogram are prognostic of outcome or not. Coming from non-native English speakers, the quality of English is alright but needs improvement.

Authors find that other than ER (which is not a variable in the MSKCC nomogram), none of the variables assessed have a significant association with the recurrence risk. (The most robust study on prognostic role of ER in DCIS is Thorat et al, PMID 33727261 – merits citation)

A small single-institution series with a small number of events is rightly described as a limitation for the findings. However, the key question is “how does one validate a nomogram?”. This reviewer is of the opinion that merely investigating whether individual variables in a nomogram are associated with the outcome of interest is not a sufficiently robust approach since a nomogram is a specific multivariate model with precisely identified beta coefficients and therefore the whole model (and not it’s individual parts) needs to be evaluated (see Sweldens et al). From the description in the manuscript, this does not appear to be the case and therefore a re-analysis is necessary. Evaluating C-index and concordance probability estimate (CPE) would be sufficient.

NOTE: This reviewer agrees with the authors’ statement that MSKCC nomogram lacks important molecular markers as its major weakness and therefore probably not a very strong and reproducible prognostic tool. However, appropriate statistical procedures still need to be followed before stating that conclusion.

Table 2: Are these multivariate analyses?

It is best to present both, univariate and multivariate ORs in the same table.

Furthermore, include RT as a variable.

Also recommend presenting (2 multivariate) models, one with and second without RT.

Minor –

Hormonotherapy – should be written as adjuvant endocrine therapy

NSABP-B24 reference – Author name not correctly formatted – it should be Allred C

Author Response

Revisor 1

Authors interrogated their single-institution database of DCIS patients treated with breast conserving surgery to see if variables in the MSKCC nomogram are prognostic of outcome or not. Coming from non-native English speakers, the quality of English is alright but needs improvement.

Authors response: It has been corrected again by a Native English speaker.

Authors find that other than ER (which is not a variable in the MSKCC nomogram), none of the variables assessed have a significant association with the recurrence risk. (The most robust study on prognostic role of ER in DCIS is Thorat et al, PMID 33727261 – merits citation)

Authors response: Cited as reviewers suggest

A small single-institution series with a small number of events is rightly described as a limitation for the findings. However, the key question is “how does one validate a nomogram?”. This reviewer is of the opinion that merely investigating whether individual variables in a nomogram are associated with the outcome of interest is not a sufficiently robust approach since a nomogram is a specific multivariate model with precisely identified beta coefficients and therefore the whole model (and not it’s individual parts) needs to be evaluated (see Sweldens et al). From the description in the manuscript, this does not appear to be the case and therefore a re-analysis is necessary. Evaluating C-index and concordance probability estimate (CPE) would be sufficient.

Authors response: In order to validate the MSKCC predictive model, a multivariate logistic regression using the risk factors included in the MSKCC model was performed. As the reviewer suggests, to provide a more robust approach to the external validation of the model, a resampling Bootstrap analysis obtaining a cross-validation of 1000 replications was used to internally validate the MSKCC nomogram in our patients’ cohort, providing and Area Under the Curve analysis of the predictive model.

NOTE: This reviewer agrees with the authors’ statement that MSKCC nomogram lacks important molecular markers as its major weakness and therefore probably not a very strong and reproducible prognostic tool. However, appropriate statistical procedures still need to be followed before stating that conclusion.

Table 2: Are these multivariate analyses?

It is best to present both, univariate and multivariate ORs in the same table.

Furthermore, include RT as a variable.

Authors response: As the reviewers suggest, we have added the OR for both the univariate and multivariate analysis and used RT as a variable.

Also recommend presenting (2 multivariate) models, one with and second without RT.

Authors response: The two suggested multivariate models present similar OR, without reaching statistical significance, for this reason, the authors left only the model including RT.

Minor –

Hormonotherapy – should be written as adjuvant endocrine therapy

Authors response: Corrected

NSABP-B24 reference – Author name not correctly formatted – it should be Allred C

Authors response: Corrected

Reviewer 2 Report

We thank the authors for this DCIS series and their attempt to help select patients for tailored treatment by assessing the MSKCC nomogram clinical value. This paper is well-written, easy to read, and with net deductions.

I have just one comment. The authors should shorten the patient characteristics description as they repeat what can be extracted from table 1. Only complementary comments or some explanation should be added.

Author Response

Revisor 2

Comments and Suggestions for Authors

We thank the authors for this DCIS series and their attempt to help select patients for tailored treatment by assessing the MSKCC nomogram clinical value. This paper is well-written, easy to read, and with net deductions.

I have just one comment. The authors should shorten the patient characteristics description as they repeat what can be extracted from table 1. Only complementary comments or some explanation should be added.

Authors response: Corrected